# Erythritol Ameliorates Small Intestinal Inflammation Induced by High-Fat Diets and Improves Glucose Tolerance

**DOI:** 10.3390/ijms22115558

**Published:** 2021-05-24

**Authors:** Rena Kawano, Takuro Okamura, Yoshitaka Hashimoto, Saori Majima, Takafumi Senmaru, Emi Ushigome, Mai Asano, Masahiro Yamazaki, Hiroshi Takakuwa, Ryoichi Sasano, Naoko Nakanishi, Masahide Hamaguchi, Michiaki Fukui

**Affiliations:** 1Department of Endocrinology and Metabolism, Graduate School of Medical Science, Kyoto Prefectural University of Medicine, Kyoto 602-8566, Japan; rena0421@koto.kpu-m.ac.jp (R.K.); d04sm012@koto.kpu-m.ac.jp (T.O.); y-hashi@koto.kpu-m.ac.jp (Y.H.); saori-m@koto.kpu-m.ac.jp (S.M.); semmarut@koto.kpu-m.ac.jp (T.S.); emis@koto.kpu-m.ac.jp (E.U.); maias@koto.kpu-m.ac.jp (M.A.); masahiro@koto.kpu-m.ac.jp (M.Y.); naoko-n@koto.kpu-m.ac.jp (N.N.); michiaki@koto.kpu-m.ac.jp (M.F.); 2Agilent Technologies, Chromatography Mass Spectrometry Sales Department, Life Science and Applied Markets Group, Tokyo 192-8510, Japan; hiroshi_takakuwa@agilent.com; 3AiSTI Science Co., Ltd., Wakayama 640-8341, Japan; sasano@aisti.co.jp

**Keywords:** erythritol, short-chain fatty acid, innate lymphoid cells, ILC3

## Abstract

Background: Erythritol, a sugar alcohol, is widely used as a substitute for sugar in diets for patients with diabetes or obesity. Methods: In this study, we aimed to investigate the effects of erythritol on metabolic disorders induced by a high-fat diet in C57BL/6J mice, while focusing on changes in innate immunity. Results: Mice that were fed a high-fat diet and administered water containing 5% erythritol (Ery group) had markedly lower body weight, improved glucose tolerance, and markedly higher energy expenditure than the control mice (Ctrl group) (*n* = 6). Furthermore, compared with the Ctrl group, the Ery group had lesser fat deposition in the liver, smaller adipocytes, and significantly better inflammatory findings in the small intestine. The concentrations of short-chain fatty acids (SCFAs), such as acetic acid, propanoic acid, and butanoic acid, in the serum, feces, and white adipose tissue of the Ery group were markedly higher than those in the Ctrl group. In flow cytometry experiments, group 3 innate lymphoid cell (ILC3) counts in the lamina propria of the small intestine and ILC2 counts in the white adipose tissue of the Ery group were markedly higher than those in the Ctrl group. Quantitative real-time reverse transcription polymerase chain reaction analyses showed that the Il-22 expression in the small intestine of the Ery group was markedly higher than that in the Ctrl group. Conclusions: Erythritol markedly decreased metabolic disorders such as diet-induced obesity, glucose intolerance, dyslipidemia, and fat accumulation in the mouse liver by increasing SCFAs and modulating innate immunity.

## 1. Introduction

Sugar alcohols are not completely absorbed into the bloodstream and thus lead to a smaller increase in blood sugar levels than “regular” sugar. Therefore, in recent years, they have been widely used as an alternative to sugar, as part of dietary treatments for metabolic syndrome, as well as type 2 diabetes and obesity. Sugar alcohols are classified as indigestible carbohydrates [1,2]. Digestible carbohydrates are digested in the small intestine and play an important role as an energy source for the body, while indigestible carbohydrates are metabolized during fermentation in the lower gastrointestinal tract by gut microbiota and are characterized by their beneficial physiological effects on human health, such as improvement of gut microbiota, improvement of fecal properties, reduction in energy intake, suppression of elevated blood glucose and insulin secretion, inhibition of toxic substances and toxic odorants derived from intestinal bacteria, and improvement in lipid metabolism [3,4,5].

Most sugar alcohols are produced by reducing the carbonyl group of monosaccharide, disaccharide, or oligosaccharide to an alcohol group, but erythritol is produced by yeast fermentation of glucose [6]. Sugar alcohols are produced by adding hydrogen to carbohydrates; therefore, unlike aspartame and saccharin, they are not considered artificial sweeteners. Erythritol is a four-carbohydrate alcohol with a sweetness that is approximately 75% that of sugar (sucrose), has low hygroscopicity, produces a cooling sensation, and crystallizes easily. Erythritol is widely distributed in the plant kingdom, including in mushrooms and fruits, and is also found in high concentrations in fermented foods such as sake, wine, miso, and soy sauce [6]. When erythritol is orally administered to rats, more than 90% is excreted in the urine without being metabolized and less than 10% is excreted as carbon dioxide [7,8]. Since the erythritol absorbed is not utilized by the tissues, it is excreted in the urine. Unabsorbed erythritol is transferred to the large intestine, where it undergoes fermentation by intestinal microorganisms and is converted to short-chain fatty acids (SCFAs). Increase in the erythritol dose leads to increase in the ratio of unabsorbed erythritol transferred to the large intestine and in the ratio of excretion as exhaled carbon dioxide [7,8]. Blood glucose levels do not increase when erythritol is ingested by healthy animals; therefore, the blood insulin concentration is also unaffected [7,8]. A previous animal study demonstrated that the administration of erythritol improved fasting blood glucose levels and HOMA-IR [9].

In the last decade, innate lymphocytes (ILCs), a group of lymphocytes that are involved in innate immunity and do not express antigen receptors, were discovered. ILCs are now classified into three groups: ILC1, ILC2, and ILC3. ILC1 produces interferon (IFN)-γ and protects against intracellular bacteria and viruses by activating macrophages. ILC1 differentiation is induced by the transcription factor T-bet, which is involved in inducing ILC1 differentiation (type 1 immune response). Conversely, ILC2s inhibit the tumorigenicity (ST)2 receptor and secrete type 2 cytokines such as interleukin (IL)-5 and IL-13 in response to cytokines such as IL-33 [10,11]. The transcription factor GATA3 is involved in the induction of ILC2 differentiation [12]. Moreover, ILC2s exhibit plasticity and differentiate into IL-12–secreting ILC1-like cells, that is, the so-called ex-ILC2s, upon stimulation by IL-1β [13]. In obese mice, ILC2s decrease in adipose tissue [14]. Galle-Treger et al. [15] showed that the ILC2s in adipose tissue improve glucose homeostasis via glucocorticoid-induced tumor necrosis factor receptors. Taken together, these studies indicate that ILC2s may play a protective role in the pathogenesis of obesity. ILC3 produces IL-17 and IL-22, mobilizes neutrophils to defend against extracellular bacteria and fungi, and is involved in epithelial cell activation and proliferation, and in the liver, prevents the development of nonalcoholic fatty liver disease (NAFLD) and nonalcoholic steatohepatitis (NASH) [16]. As well as ILC2, ILC3s exhibit plasticity and the function of ILC3s is altered by the expression of the transcription factors RORγt and T-bet [17]. On stimulation with cytokines such as IL-12 and IL-18, ex-RORγt-positive ILC3s with T-bet-positive characteristics, that is, T-bet-positive ILC3s, increase and RORγt-positive ILC3s decrease, indicating that ILC3s can respond to environmental cues. A previous study showed that T-bet-positive ILC3s can produce IFN-γ and suppress IL-17 and IL-22 production [18]. Thus, T-bet-positive ILC3s exert a function similar to that of ILC1.

On the other hand, in a previous report, it was shown that intestinal immune cells remotely control chronic inflammation in adipose tissue and modulate systemic insulin sensitivity in obese mice fed a high-fat diet [19], which suggest that the link between intestinal and adipose tissue immunity plays a key role in the pathogenesis of obesity.

Recently, ILCs were found to express receptors for SCFAs, such as G protein–coupled receptor (GPR) 41 (also known as free fatty acid receptor (FFAR3) and GPR43 (FFAR2), that have been reported to stimulate the activation of phosphatidylinositol-3 kinase (PI3K), signal transducer and activator of transcription 3 (Stat3), Stat5, and mechanistic target of rapamycin (mTOR), which are important for ILC proliferation [20]. Therefore, we hypothesized that the ameliorating effect of erythritol on various metabolic disorders may be related to the anti-inflammatory effect of innate immunity due to increased SCFA production in the intestine.

In this study, we focus on SCFA and FFAR2 and FFAR3, with a particular focus on eWAT, which strongly expresses FFAR2, and erythritol was administered to experimental animals to evaluate its protective effect against metabolic disorders caused by a high-fat diet (HFD), while focusing on intestinal innate immunity.

## 2. Results

### 2.1. Erythritol Decreased Obesity and Glucose Intolerance Induced by the HFD

From the age of 11 weeks, the body weight of mice fed an HFD and erythritol (Ery group) were significantly lower than those of mice fed an HFD (Ctrl group) (Figure 1A). In contrast, the cumulative oral intake and water consumption did not significantly differ between both groups (Figure 1B,C). In the iPGTT and ITT, the AUCs of blood glucose in the Ery group were significantly lower than those in the Ctrl group (*p <* 0.001 for both; Figure 1D–G).

### 2.2. Erythritol Enhanced Energy Metabolism

To evaluate the increase in the energy metabolism of erythritol, energy metabolism was measured by housing 20-week-old mice in metabolic cages. O_2_ consumption and CO_2_ content, which indicate energy expenditure, increased in the Ery group (Figure 1H–K).

### 2.3. Liver and Epididymal Fat Weights Did Not Significantly Differ between the Two Groups

The liver and epididymal fat weights in the Ery group did not significantly differ, compared with those in the Ctrl group (Figure 2A–D).

### 2.4. Erythritol Reduced Levels of Hepatic Enzyme and Serum Lipids Induced by the HFD

In the blood biochemistry analysis, the serum aspartate transaminase (AST), alanine aminotransferase (ALT), triglyceride (TG), total cholesterol, low-density lipoprotein (LDL) cholesterol, and nonesterified fatty acid (NEFA) levels of the Ery group were significantly lower than those of the Ctrl group (AST: *p* = 0.020, ALT: *p* = 0.012, TG: *p* = 0.035, total cholesterol: *p* = 0.046, LDL cholesterol: *p* = 0.006, NEFA: *p* = 0.007). In contrast, the high-density lipoprotein (HDL) cholesterol levels in the Ery group were higher than those in the Ctrl group (HDL: *p* = 0.034; Figure 2E–K).

### 2.5. Erythritol Ameliorated the NAFLD, Increase in Adipocyte Size, and Small Intestinal Inflammation Induced by the HFD

The Ery group had significantly lesser hepatic fat accumulation (Figure 3A,B,D) and liver fibrosis (Figure 3A,C) than the Ctrl group. In addition, histological analysis of the eWAT revealed that the average cell area and lipid droplet area per cell in the Ery group were significantly lower than those in the Ctrl group (Figure 3A,E,F).

Histological analysis of the small intestine revealed that the villus height and width in the Ery group were higher than those in the Ctrl group (Figure 3A,G,H). Conversely, the crypt depth in the Ery group was lower than that in the Ctrl group (Figure 3I).

### 2.6. Erythritol Increased SCFAs in the Serum, Feces, and eWAT

We investigated the concentrations of three SCFAs, that is, acetic acid, propanoic acid, and butanoic acid, in the serum, feces, and eWAT. The levels of all three SCFAs in the Ery group were significantly higher than those in the Ctrl group (Figure 4A–I).

### 2.7. Erythritol Increased ILC2s in the eWAT and ILC3 in the Lamina Propria of the Small Intestine

Changes in the numbers of ILCs and macrophages in the eWAT and lamina propria of the small intestine were examined using flow cytometry (Appendix A). The ratios of ILC1s in CD45 positive cells, which have been reported to promote adipose tissue fibrosis and diabetes [21] in the eWAT of the Ery group, were significantly lower than those in the eWAT of the Ctrl group (*p* = 0.001; Figure 5A), whereas the ratios of ILC2s in CD45 positive cells, which have been reported to promote beiging of WAT and suppress obesity [13] in the Ery group, were significantly higher than those in the Ctrl group (*p* = 0.029; Figure 5B).

The M1 pro-inflammatory macrophages and M2 anti-inflammatory macrophages [22,23] were also evaluated. Compared with the Ctrl group, the Ery group showed a lower ratio of M1 macrophages and higher ratio of M2 macrophages in CD45 positive cells in the eWAT (*p <* 0.0001; Figure 5C,D).

We also evaluated ILC1-like cells, the so-called ex-ILC2s, which exhibit plasticity and secrete IL-12 on stimulation by IL-1β [24]. The ratio of ex-ILC2s in CD45 positive cells in the eWAT significantly decreased in the Ery group (*p* = 0.049; Figure 5E). The ratio of ILC1s in CD45 positive cells in the lamina propria of the small intestine in the Ery group was significantly lower than that in the Ctrl group (*p <* 0.0001; Figure 5F). We also measured ILC3s, which produce IL-22, mobilize neutrophils for defense against extracellular bacteria and fungi, and are also involved in the activation and proliferation of epithelial cells [25]. The ratio of ILC3s in CD45 positive cells in the lamina propria of the small intestine in the Ery group was significantly higher than that in the Ctrl group (*p <* 0.0001; Figure 5G). Moreover, compared with the Ctrl group, the Ery group showed a decreased ratio of M1 macrophages and an increased ratio of M2 macrophages in CD45 positive cells in the lamina propria of the small intestine (M1 macrophages: *p* = 0.030, M2 macrophages: *p <* 0.0001; Figure 5H,I). Additionally, the ratio of T-bet-positive ILC3s in CD45 positive cells, which secrete high amounts of IFN-γ [8], decreased in the Ery group (*p* = 0.009; Figure 5J).

### 2.8. Erythritol Decreased the Expression of Genes Related to Inflammation, Glucose Transporters, and Fatty Acid Transporters in the Small Intestine

We investigated gene expression in the small intestine. The expression of Il-22, a cytokine that is secreted by ILC3 and induces the expression of mucin genes in mucosal epithelial cells via STAT3-dependent signaling [25], was significantly elevated in the Ery group (*p* = 0.0003; Figure 6A). The expression of genes related to inflammation, such as Il6, Il1b, and Il23a, in the Ery group was significantly lower than that in the Ctrl group (Il6: *p* = 0.043, Il1b: *p* = 0.009, and Il23a: *p* = 0.021; Figure 6B–D). Moreover, the expression of Sglt1, a glucose transporter, in the Ery group was lower than that in the Ctrl group (*p* = 0.030; Figure 6E), and the expression of Cd36, a long-chain fatty acid, in the Ery group was lower than that in the Ctrl group (*p* = 0.013; Figure 6F).

### 2.9. Erythritol Increased the Expression of Genes Encoding Cytokines Secreted by ILC2s and Decreased the Expression of Inflammation-Related Genes in the eWAT

The expression of *Il-5* and *Il-13*, which are cytokines secreted by ILC2s, in the eWAT was significantly higher in the Ery group than in the Ctrl group (*Il5*: *p* = 0.025, *Il13*: *p* = 0.003; (Figure 6G,H), whereas the expression of Il-33, which activates ILC2, in the Ery group was significantly lower than that in the Ctrl group (*p* = 0.026; Figure 6I). The expression of inflammation-related genes, that is, *Il1b*, *Tnfa*, and *Il6*, in the Ery group was significantly lower than that in the Ctrl group (*Il1b*: *p* = 0.0005, *Tnfa*: *p* = 0.041, and *Il6*: *p* = 0.0001; Figure 6J,K).

### 2.10. Erythritol Decreased Ffar2 and Ffar3 Expression in the Colon and Increased it in the eWAT

We investigated the expression of SCFA receptors, such as *Ffar2* and *Ffar3*. The *Ffar2* and *Ffar3* expression in the colon of the Ery group was significantly lower than that in the colon of the Ctrl group (*Ffar2*: *p* = 0.020, *Ffar3*: *p* = 0.0008; Figure 7A,B). In contrast, the *Ffar2* and *Ffar3* expression in the eWAT of the Ery group was significantly higher than that in the eWAT of the Ctrl group (*Ffar2*: *p* = 0.023, *Ffar3*: *p* = 0.022; Figure 7C,D).

## 3. Discussion

In this study, we found that erythritol significantly decreased metabolic disorders, including obesity, glucose intolerance, dyslipidemia, and fat accumulation in the liver, induced by an HFD. Moreover, erythritol administration increased SCFAs in the feces, which is similar to previous findings [8], and in the serum and WAT. On the other hand, liver and epididymal fat weights tended to be lower in the erythritol group, although the differences were not statistically significant. However, the administration of erythritol had metabolic effects, such as reduced fat accumulation in the liver and improved inflammation in visceral fat, which suggested that subcutaneous fat was also reduced, although not measured, resulting in body weight diminution.

Erythritol is produced by yeast fermentation of glucose. More than 90% of orally ingested erythritol is excreted in the urine without being metabolized, and less than 10% is excreted as carbon dioxide [7,8]. Unabsorbed erythritol moves to the large intestine, where it undergoes fermentation by the gut microbiota and is converted to SCFAs. The proportion of unabsorbed erythritol transferred to the large intestine increases with increase in dosage, and the proportion excreted as carbon dioxide in the exhaled air also increases. This study also confirmed that SCFA production increased in the intestine, which may be due to erythritol fermentation by the gut microbiota. Gut microbiota use energy sources such as glucose and fatty acids that are directly obtained from food and also produce SCFAs, such as acetic acid, butanoic acid, and propanoic acid, as an energy source [26]. The FFAR, a G protein–coupled receptor (GPR), is a receptor for these SCFAs. FFAR2, FFAR3, and GPR109A are expressed in white adipose tissue, which inhibits lipolysis and enhances leptin secretion [27]. FFAR2 is mainly activated by acetic acid and propanoic acid [28], while FFAR3 is mainly activated by propanoic acid and butanoic acid [29] and has been shown to be completely dependent on the SCFA concentration in the circulating blood. Kimura et al. [28] reported that when mice were fed an HFD, the volume of WAT in transgenic mice with high FFAR2 expression was lower than that in wild-type mice, and stimulation of FFAR2 suppressed insulin signaling in adipocytes because of inhibition of Akt phosphorylation. In addition, in other previous studies, the expression of FFAR2 and FFAR3 in visceral fat of mice fed a high-fat diet was significantly lower than that of mice fed a normal diet [30]. Taken together, we hypothesized that the expression of FFAR2 and FFAR3 was significantly increased in high-fat diet-fed mice in a compensatory manner to improve insulin resistance in visceral fat. In this study, erythritol administration significantly increased FFAR2 and FFAR3 expression in the WAT. This suggests that erythritol improved insulin resistance in visceral fat caused by HFD.

In this study, *Pparg* mRNA expression, which is involved in the differentiation of adipocytes in the WAT of the Ery group significantly increased, compared to that of Ctrl group. In the Ery group, the adipocytes became smaller, indicating genetic changes. In a previous report, it was reported that mice with adipose tissue-specific knockout (KO) of Pparγ did not show any effect of short-chain fatty acid administration, such as weight loss or increase in insulin sensitivity, indicating that SCFA may be a selective PPARγ modulator [31]. In the present study, erythritol administration increased the concentration of short-chain fatty acids in WAT, which may have led to adipocyte miniaturization through increased *Pparg* mRNA expression.

Additionally, energy expenditure, as evaluated by indirect calorimetry, significantly increased with the erythritol administration. In a previous study, FFAR2 was shown to be an important regulator of inflammation in a mouse model of colitis [32]. Moreover, Mosińska et al. [33] reported that use of an HFD led to higher FFAR2 expression in the colon than a normal diet did, which is in line with our results. This finding is thought to be due to increased gene expression of FFAR2, which suppresses inflammation caused by HFDs in the colon. Furthermore, Lu et al. [30]. reported that feeding an HFD increased the gene expression of FFAR2 and FFAR3 in the colon, compared to feeding an ND and the administration of short chain fatty acid decreased the expression of FFAR2 and FFAR3 in the colon. In addition, a positive correlation with the expression of PYY and GLP-1 has been reported in the same report, which suggested that high expression of FFAR2 and FFAR3 in the colon might prevent weight gain by increasing PYY and GLP-1 expression and suppressing food intake. In this study, erythritol-induced increase of short-chain fatty acids in the colon may have decreased the expression of FFAR2 and FFAR3 in the colon. Whether the health effects of erythritol are SCFA-dependent will require further validation in FFAR KO mice.

In the current study, erythritol significantly decreased SGLT-1 expression in the small intestine. A previous study showed that HFD-induced dysbiosis is associated with GLP-1 resistance [34]. The coabsorption of Na+/glucose in the intestine by SGLT1 and the release of GLP-1 from enteroendocrine L cells have been reported to occur simultaneously [35]. In the Ery group, wherein the dysbiosis decreased, the SGLT-1 expression also decreased, suggesting that glucose absorption from the small intestine was suppressed and glucose tolerance may have improved. As with FFAR, this finding needs to be tested in KO mice to determine whether erythritol has a glycemic effect via changes in SGLT-1 expression.

In the current study, erythritol significantly decreased CD36 expression in the small intestine. CD36-deficient mice have been previously found to exhibit worsened small intestinal inflammation [36], suggesting that HFD-induced small intestinal inflammation compensatively increases CD36 expression, while erythritol-induced improvement of small intestinal inflammation decreases CD36 expression. Furthermore, erythritol treatment reduced Cd36-mediated absorption of saturated fatty acids, which decreased intrahepatic fat accumulation, significantly improved lipid profiles such as cholesterol, TG, and NEFA, and significantly decreased gene expression such as Scd1 and Fasn, which are involved in fatty acid synthesis in the liver. It was also suggested that the expression of genes such as Scd1 and Fasn involved in fatty acid synthesis in the liver was also significantly decreased.

FFAR signaling has been reported to play an important role in innate immunity. Chun et al. [37] reported that SCFAs stimulate the FFAR2 of ILC3s in the intestine and promote the secretion of IL-22 via Akt and Stat3 signaling. The cytokine IL-22 is induced in immune cells upon infection with pathogenic bacteria [25]. It acts on intestinal epithelial cells that express the IL-22 receptor in the intestine and induces the production of mucus and antimicrobial proteins in intestinal epithelial cells and fucosylation of the epithelial cell surface, thereby enhancing the defense against bacterial infection. Moreover, IL-22 has been reported to reduce endoplasmic reticulum/oxidative stress, improve the integrity of the mucosal barrier, and reverse intestinal epithelial cell stress and inflammation induced by HFDs [38].

In the Ery group, the percentage of ILC1s in the small intestine decreased and the percentage of RORγt-positive ILC3s increased. ILCs exhibit functional plasticity in response to their environment, and the function of ILC3s is altered by the expression of the transcription factors RORγt and T-bet [17]. On stimulation with cytokines such as IL-12 and IL-18, ex-RORγt-positive ILC3s with T-bet-positive characteristics, that is, T-bet-positive ILC3s, increase and RORγt-positive ILC3s decrease, indicating that ILC3s can respond to environmental cues. A previous study showed that T-bet-positive ILC3s can produce IFN-γ and suppress IL-17 and IL-22 production [18]. Thus, T-bet-positive ILC3s exert a function similar to that of ILC1. In the current study, the ILC3s in the lamina propria of the small intestine and the IL-22 expression increased and the intestinal epithelial cell inflammation decreased in the mice administered erythritol. Moreover, T-bet-positive ILC3s significantly decreased on erythritol administration. Taken together, the SCFAs increased in the intestine on administration of erythritol, which was absorbed into the intestine and stimulated FFAR2, which in turn activated ILC3 and enhanced IL-22 secretion, thereby suppressing intestinal inflammation.

ILC2s exhibit plasticity and have been reported to differentiate into IL-12-secreting ILC1-like cells, the so-called ex-ILC2s, upon stimulation by IL-1β [13]. The ILC2s in adipose tissue have been reported to decrease in obese mice [39] and improve glucose homeostasis via the glucocorticoid-induced tumor necrosis factor receptor [15]; they are prevalent in healthy adipose tissue, wherein they contribute to adipose tissue remodeling, counteracting the inflammatory effect of obesity and inducing browning of white adipose tissue [40,41]. Taken together, these studies suggest that ILC2s play a protective role in the pathogenesis of obesity. In the current study, erythritol administration increased ILC2 counts and decreased ex-ILCs in the WAT. Moreover, erythritol decreased the expression of inflammation-related genes in the WAT, which might contribute to improvement in insulin resistance.

On the other hand, in a human study, young adults with incident central adiposity gain had statistically significantly higher blood erythritol levels [42]. However, the authors note that this may involve endogenous production of erythritol from glucose via the pentose-phosphate pathway, which does not imply that erythritol is involved in the worsening of obesity.

A limitation of this study is the lack of a third group using another substitute of sugar to demonstrate that the effects here observed are erythritol specific. Moreover, we did not have the data of insulin in the iPGTT and ITT.

## 4. Materials and Methods

### 4.1. Animals

All experimental procedures were approved by the Committee for Animal Research at the Kyoto Prefectural University of Medicine (M2020-42). Since the sexual cycle of female mice has been reported to affect innate immunity [43], seven-week-old C57BL/6J (wild-type) male mice were purchased from Shimizu Laboratory Supplies (Kyoto, Japan), littermate mice that were born at the same time at a mouse supply facility. The mice were housed in a specific pathogen-free controlled environment, divided into weight-matched two groups and cages in the same rack at our facility, and kept in the following two groups of six mice per cage. The mice were maintained in an environmentally controlled room (23 ± 1.5 °C) with a 12-h light/12-h dark cycle (7 a.m.–7 p.m.). The mice were fed an HFD (459 kcal/100 g, 20% protein, 20% carbohydrate, and 60% fat (lard); D12492; Research Diets Inc., New Brunswick, NJ, USA) for 12 weeks, starting at 8 weeks of age (Ctrl group). In addition, the erythritol group (Ery group) was administered water containing 5% erythritol. The body weights of the mice were measured every week. When the mice were 20 weeks old, they were sacrificed by administering a combination anesthetic comprising 0.3 mg/kg medetomidine, 4.0 mg/kg midazolam, and 5.0 mg/kg butorphanol after an overnight fast [15].

### 4.2. Measurement of Oral Intake and Water Consumption

The cumulative oral intake was measured for 12 weeks, and water consumption of the mice were measured at 20 weeks. The weighed food was placed in a trough, and the measured water was placed in a bottle in each cage. After 24 h, the amounts of food and water remaining were measured.

### 4.3. Intraperitoneal Glucose Tolerance Test (iPGTT) and Insulin Tolerance Test (ITT)

An iPGTT (2 g/kg body weight) was performed for 20-week-old mice that had been fasted for 16 h (6 animals per group, three days before sacrifice). In addition, after a 5 h fast, the mice were treated with 0.5 U/kg body weight insulin for the ITT (two days before sacrifice). Blood glucose levels were measured in drops of blood obtained at the time points, 0 min, 30 min, 60 min, and 120 min after injection for iPGTT and 0 min, 15 min, 30 min, 60 min, and 120 min after injection for ITT, by using a glucometer (Gultest Neo Alpha; Sanwa Kagaku Kenkyusho, Nagoya, Japan). The iPGTT and ITT results were analyzed by measuring the area under the curve (AUC). The O_2_/CO_2_ analyzer we used in this study measures the VO_2_ and VCO_2_ in the cage every 3 min, which can obtain 20 pieces of data per hour. In each group of 6 animals, the mean and SD values of the light and dark periods were calculated.

### 4.4. Indirect Calorimetry

In vivo indirect open-circuit calorimetry was performed in metabolic chambers at a controlled ambient temperature (24 °C ± 2 °C). A constant air flow (0.6 L/min) was drawn through the chamber and monitored using a metabolic analyzer (O_2_/CO_2_ Analyzer MM202R; Muromachi Kikai Co., Ltd., Tokyo, Japan). The oxygen consumption rate (VO_2_) and carbon dioxide production rate (VCO_2_) were measured for 48 h during 12 h light/12 h dark cycles at the inlets and outlets of the sealed chambers. Throughout these experiments, the mice had ad libitum access to food and water.

### 4.5. Biochemistry

In designated experiments, mice were fed the HFD from 8 to 20 weeks of age. Mice were sacrificed after an overnight fast. Blood samples were taken from fasted mice and alanine aminotransferase (ALT) levels, total cholesterol, triglycerides, and non-esterified fatty acids (NEFA) were measured. The biochemical examinations were performed at FUJIFILM Wako Pure 18 Chemical Corporation (Osaka, Japan).

### 4.6. Liver Histological Analysis

Liver tissue was obtained and was fixed with 10% buffered formaldehyde or embedded in paraffin. Tissue sections were prepared and stained with hematoxylin and eosin (H&E) or Masson’s trichrome stain.

The liver sections were also subjected to Oil Red O staining. The tissues were fixed in 4% paraformaldehyde overnight at 4 °C. The liver tissues were frozen in OCT compounds, cut into 4 μm thick sections, mounted onto slides, and allowed to dry for 1–2 h. The sections were then rinsed with phosphate-buffered saline (PBS; pH = 7.4). After the slides were air dried, they were placed in 100% propylene glycol for 2 min and then stained with 0.5% Oil Red O solution in propylene glycol for 30 min. The slides were subsequently transferred to an 85% propylene glycol solution for 1 min, rinsed in distilled water twice, and processed for hematoxylin counterstaining.

To evaluate the severity of nonalcoholic fatty liver disease (NAFLD), the NAFLD activity score (NAS) [44], a well-known standard for assessing nonalcoholic steatohepatitis (NASH) severity and measuring changes in NAFLD, was determined. The NAS was evaluated by a trained hepatopathologist while masking the experimental conditions [11,19]. Briefly, the scoring system consisted of 14 histological features, four of which were evaluated semiquantitatively: hepatocellular ballooning (0–2), lobular inflammation (0–2), steatosis (0–3), and fibrosis (0–4). In addition, to assess fibrosis, stage 1 was classified as follows: 1A, mild pericentral perisinusoidal fibrosis; 1B, moderate or greater perisinusoidal fibrosis; and 1C, fibrosis in the portal region or periportal vein. Perisinusoidal or periportal fibrosis was classified as stage 2, bridging fibrosis as stage 3, and liver cirrhosis as stage 4 [12,45].

Images were captured with the BZ-X710 fluorescence microscope (Keyence, Osaka, Japan), followed by analysis of the Oil Red O-stained area of the liver tissue by using the ImageJ software (version 1.52u, Java 1.8 (Wayen Rasband, U.S. National Institutes of Health; Bethesda, MD, USA).

### 4.7. White Adipose Tissue Histology

Epididymal white adipose tissue (eWAT) was obtained and fixed with 10% buffered formaldehyde or embedded in paraffin. Adipose tissue sections were prepared and stained with H&E. The average cell area and lipid droplets per cell were measured using the ImageJ software according to the method described in a previous study [46].

### 4.8. Small Intestine Histological Analysis

Small intestinal tissue samples were obtained and either fixed with 10% buffered formaldehyde or embedded in paraffin. Tissue sections were prepared and stained with H&E. Images were captured using a fluorescence microscope (BZ-X710). The height/width of the villus and crypt depth were analyzed using the ImageJ software.

### 4.9. Measurement of SCFA Concentrations in Feces and Serum Samples

The SCFA composition of the murine rectal feces and serum samples was analyzed using gas chromatography (GC)—mass spectrometry (MS) on an Agilent 7890B/7000D System (Agilent Technologies, Santa Clara, CA, USA). Rectal feces (20 mg), eWAT (20 mg), and serum (50 µL) samples were added to 500 μL acetonitrile and 500 μL diluted water and ground in a ball mill at 4000× rpm for 2 min. Next, the samples were shaken at 1000× rpm for 30 min at 37 °C and centrifuged at 14,000× rpm for 3 min at room temperature. The supernatant (500 μL) was separated, added to 500 μL acetonitrile, and further shaken at 1000× rpm for 3 min at 37 °C. The samples were then centrifuged at room temperature for 3 min at 14,000× rpm, and the pH was adjusted to 8 with 0.1 mol/L NaOH to extract the SCFAs.

The SCFA concentration was then automatically determined by GC/MS using the online solid-phase extraction (SPE) method. In the SPE-GC system SGI-M100 (AiSTI Science, Wakayama, Japan), SPE and injection into the GC/MS system are automatically performed after the sample is added to the vial and set on the autosampler tray. Flash-SPE ACXs (AiSTI Science) were used for solid stratification. Fifty-microliter aliquots of each of the aforementioned sample extracts were obtained, loaded onto the solid phase, and washed with acetonitrile and water (1:1). Next, the samples were dehydrated with acetone, impregnated with 4 μL *N*-tert-butyldimethylsilyl-*N*-methyltrifluoroacetamide (MTBSTFA)-toluene solution (1:3), and eluted with hexane after derivatization on the solid phase. The final product was injected using a programmed temperature vaporizer (PTV) injector, LVI-S250 (AiSTI SCIENCE), whose temperature was maintained at 150 °C for 0.5 min, increased gradually from 25 °C/min to 290 °C, and then maintained there for 16 min. The samples were loaded onto a capillary column, Vf-5ms (30 m × 0.25 mm (inner diameter0 × 0.25 μm (membrane thickness); Agilent Technologies). The column temperature was maintained at 60 °C for 3 min, increased gradually by 10 °C/min to 100 °C and subsequently by 20 °C/min to 310 °C, and then maintained at 310 °C for 7 min. The sample was injected in split mode with a split ratio of 20:1. Each SCFA was detected in scan mode (*m*/*z*: 70–470). All results were normalized to the peak height of tetradeuteroacetic acid (0.02 nmol/μL).

### 4.10. Isolation of Mononuclear Cells from the eWAT and Small Intestines of Mice

To exclude blood contamination in the eWAT and small intestines, systemic perfusion with heparinized saline was performed before harvesting or washing the eWAT and small intestinal tissues with PBS. The eWAT was used as the representative visceral adipose tissue. To exclude blood contamination in the adipose tissue, systemic perfusion with heparinized saline was performed before harvesting or washing isolated adipose tissues with PBS. The adipose tissues were then incubated in Hanks’ balanced salt solution (HBSS; Sigma-Aldrich, St. Louis, MO, USA) containing 0.2% collagenase type 1 (Sigma-Aldrich) for 30 min at 37 °C with constant shaking. After the collagenase activity in the cell suspension was inactivated using Roswell Park Memorial Institute medium (RPMI) supplemented with 2% FCS, the cell suspension was filtered through a 40 μm nylon mesh (BD Biosciences, San Jose, CA, USA), followed by centrifugation at 300× *g* for 5 min at 4 °C. The floating adipocytes and supernatant were removed from the eWAT pellet, which was then washed and resuspended in sterilized PBS [47].

Lamina propria mononuclear cells of small intestine (mainly jejunum) were isolated using the Lamina Propria Dissociation Kit (130-097-410; Miltenyi Biotec, Germany), as per the manufacturer’s instructions. The cell pellets were resuspended in 40% Percoll^®^ (Sigma-Aldrich) and added slowly to the upper portion of the centrifuge tubes, which contained 5 mL 80% Percoll^®^ at the bottom. Lamina propria mononuclear cells were obtained by washing twice with 2% FBS/PBS and by post density gradient centrifugation at 420× *g* for 20 min at 4 °C.

### 4.11. Flow Cytometry

We used the following antibodies for gating of innate lymphoid cells: biotin-CD3e (100304; clone: 145-2C11; 1:200 dilution), biotin-CD45R/B220 (103204; clone: RA3–6B2; 1:200 dilution), biotin-Gr-1 (108404; clone: RB6-8C5; 1:200 dilution), biotin-CD11c (117304; clone: N418; 1:200 dilution), biotin-CD11b (101204; clone: M1/70; 1:200 dilution), biotin-Ter119 (116204; clone: TER-119; 1:200 dilution), biotin-FceRIa (134304; clone: MAR-1; 1:200 dilution), FITC-streptavidin (405202; 1:500 dilution), PE-Cy7-CD127 (135014; clone: A7R34; 1:100 dilution), Pacific Blue-CD45 (103116; clone: 30-F11; 1:100 dilution), PE-GATA-3 (clone TWAJ; 1:50 dilution), and APC-RORγ (clone AFKJS-9; 1:50 dilution), and Fixable Viability Dye eFluor 780 (1:400 dilution) [48]. Additionally, we used the following antibodies for gating of M1 and M2 macrophages: APC-CD45.2 (17045482; clone: 104; 1:50 dilution), PE-F4/80 (12480182; clone: BM8; 1:50 dilution), APC-Cy7-CD11b (47011282; clone: M1/70; 1:50 dilution), FITC-CD206 (MA516870; clone: MR5D3; 1:50 dilution), and PE-Cy7-CD11c (25011482; clone: N418; 1:50 dilution) [16,49,50,51] (Appendix A). Stained cells were analyzed on a FACS Canto II system and the data were analyzed using the FlowJo version 10 software (TreeStar, Ashland, OR, USA).

### 4.12. Gene Expression Levels in the Murine Small Intestinal Tissue, Colon Tissue, and eWAT

The small intestinal tissue, colon, and eWAT of fasting mice were resected and immediately frozen using liquid nitrogen. Adipose tissue was homogenized in ice-cold QIAzol Lysis reagent (Qiagen, Hilden, Germany), and total RNA was isolated according to the manufacturer’s instructions and measured using NanoDrop (Thermo Fisher Scientific). Total RNA (0.5 μg) was reverse-transcribed using a High-Capacity cDNA Reverse Transcription Kit (Applied Biosystems, Foster City, CA, USA) for first-strand cDNA synthesis by using an oligonucleotide dT primer and random hexamer priming, according to the manufacturer’s recommendations. The reverse transcription reaction was performed for 120 min at 37 °C, and inactivation of reverse transcriptase was performed for 5 min at 85 °C.

We performed real-time reverse transcription polymerase chain reaction (RT-PCR) to quantify the following: *Il22*, *Il23a*, *Sglt1*, and *Cd36* mRNA expression levels in the small intestine; *Il5*, *Il13*, *Il33*, *Il1b*, *Il6*, *Tnfa* and *Pparg* mRNA expression levels in the eWAT; *Scd1* and *Fasn* mRNA expression levels in the liver; and *Ffar2* and *Ffar3* mRNA expression levels in the colon and eWAT. RT-PCR was performed using TaqMan Fast Advanced Master Mix (Applied Biosystems), according to the manufacturer’s instructions. The following PCR conditions were used: 1 cycle of 2 min at 50 °C and 20 s at 95 °C, followed by 40 cycles of 1 s at 95 °C, and 20 s at 60 °C.

The relative expression levels of each targeted gene were normalized to the *Gapdh* threshold cycle (CT) values and quantified using the comparative threshold cycle 2^−ΔΔCT^ method, as previously described [11,52]. Signals from Ctrl group were assigned a relative value of 1.0. Samples from 6 mice from each group were examined, and RT-PCR was performed in triplicate for each sample.

### 4.13. Statistical Analysis

Differences between two groups were assessed using the unpaired *t*-test and the paired t-test for the VO_2_ and VCO_2_, respectively. We used the Prism software (version 9.0; GraphPad, San Diego, CA, USA). Differences were considered statistically significant at *p* < 0.05.

## 5. Conclusions

In summary, erythritol significantly decreased metabolic disorders, such as obesity, glucose intolerance, dyslipidemia, and fat accumulation in the liver, which were induced by an HFD. In particular, it exerted anti-inflammatory effects in the intestine via increased ILC3 and IL-22 expression in the intestine. This is a very valuable finding as there has been no previous report on the relationship between sugar alcohols and innate immunity in elucidating the mechanism of amelioration of metabolic disorders by sugar alcohols.

## Figures and Tables

**Figure 1 ijms-22-05558-f001:**
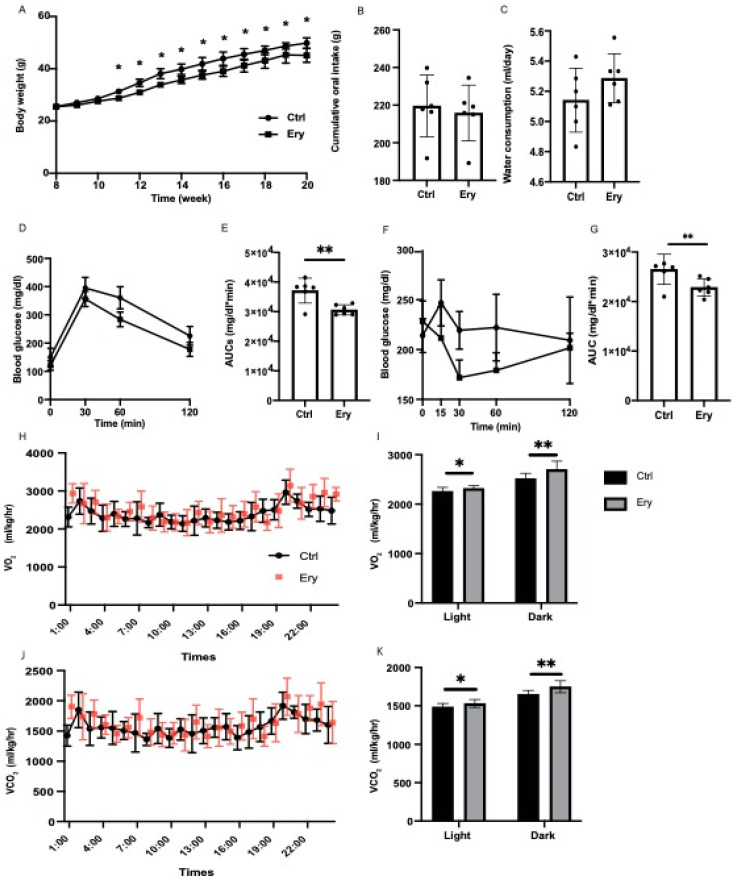
Erythritol administration significantly ameliorated impaired glucose tolerance, obesity, and metabolic disorders. (**A**) Changes in the body weights of the mice (*n* = 6). *, weight of Ery group mice was significantly lower than that of Ctrl group mice by the *t*-test. (**B**) Oral intake measured at 20 weeks of age (*n* = 6). (**C**) Water consumption measured at 20 weeks of age (*n* = 6). (**D**,**E**) When the mice were 20 weeks old, an intraperitoneal glucose tolerance test (2 g/kg body weight) was performed, followed by area under the curve analysis (*n* = 6). (**F**,**G**) When the mice were 20 weeks old, an insulin tolerance test (0.5 U/kg body weight) was performed, followed by area under the curve analysis (*n* = 6). (**H**) Real-time monitoring curve of oxygen release (VO_2_) (*n* = 6). (**I**) Quantification of O_2_ consumption (*n* = 6). (**J**) Real-time monitoring curve of carbon dioxide release (VCO_2_) (*n* = 6). (**K**) Quantification of carbon dioxide release (*n* = 6). Data have been represented in terms of mean ± SD values. * *p* < 0.05, ** *p* < 0.01, using an unpaired *t*-test and a paired t-test for the VO_2_ and VCO_2_, respectively.

**Figure 2 ijms-22-05558-f002:**
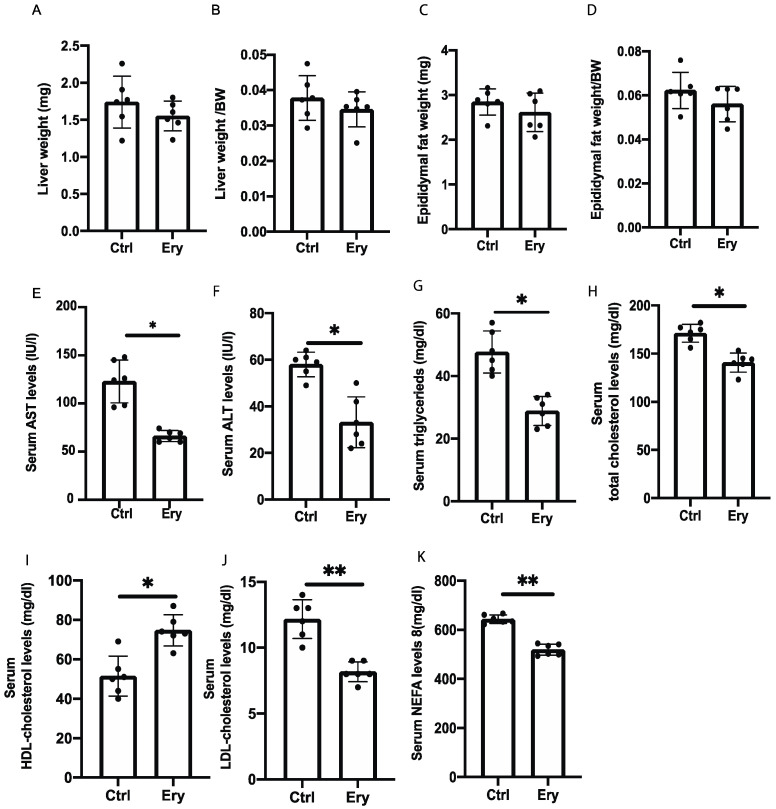
The weight of organs and blood biochemistry test results. Erythritol decreased hepatic enzyme and improved serum lipid levels. (**A**,**B**) Liver weight and liver weight:body weight ratio (*n* = 6). (**C**,**D**) Epididymal fat weight and epididymal fat weight:body weight ratio (*n* = 6). (**E**–**K**) Blood biochemistry test results for serum AST, ALT, triglyceride (TG), total cholesterol, low-density lipoprotein (LDL) cholesterol, and nonesterified fatty acid (NEFA) levels (*n* = 6). Data have been represented in terms of mean ± SD values. * *p* < 0.05 and ** *p* < 0.01, using an unpaired *t*-test.

**Figure 3 ijms-22-05558-f003:**
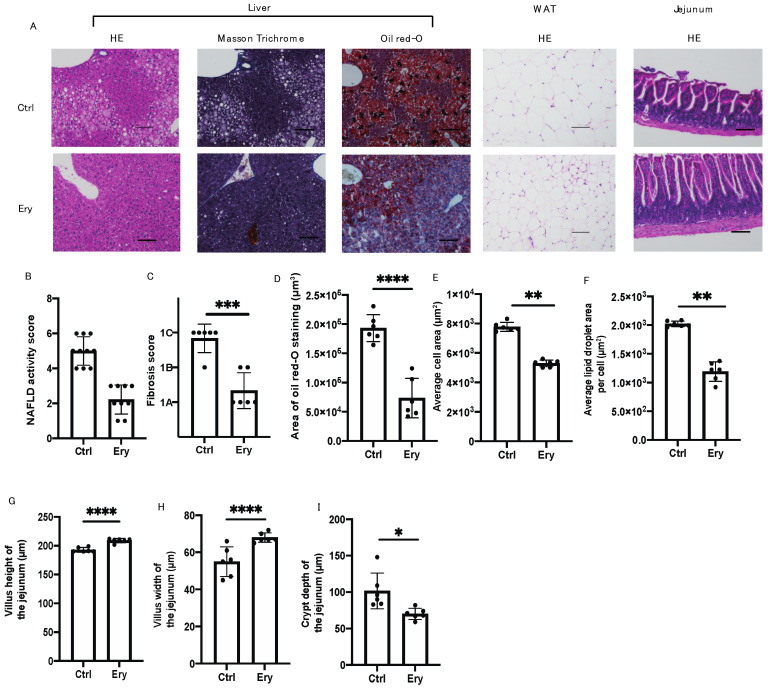
Erythritol reduced fat accumulation in the liver, liver fibrosis, adipocyte size, and small intestinal inflammation. (**A**) Representative images showing the histological features of the liver, white adipose tissue, and jejunum of the mice. (**B**) NAFLD activity score of the liver (*n* = 6). (**C**) Fibrosis score of the liver (*n* = 6). (**D**) Oil Red O–stained area of the liver (*n* = 6). (**E**) Average cell area of white adipose tissue (μm^2^) (*n* = 6). (**F**) Average lipid droplet area per cell (μm^2^) (*n* = 6). (**G**) Villus height of the jejunum (*n* = 6). (**H**) Villus width of the jejunum (*n* = 6). (**I**) Crypt depth of the jejunum (*n* = 6). Scale bar: 100 μm. Data have been represented in terms of mean ± SD values. * *p* < 0.05, ** *p* < 0.01, *** *p* < 0.001, and **** *p* < 0.0001, using an unpaired *t*-test.

**Figure 4 ijms-22-05558-f004:**
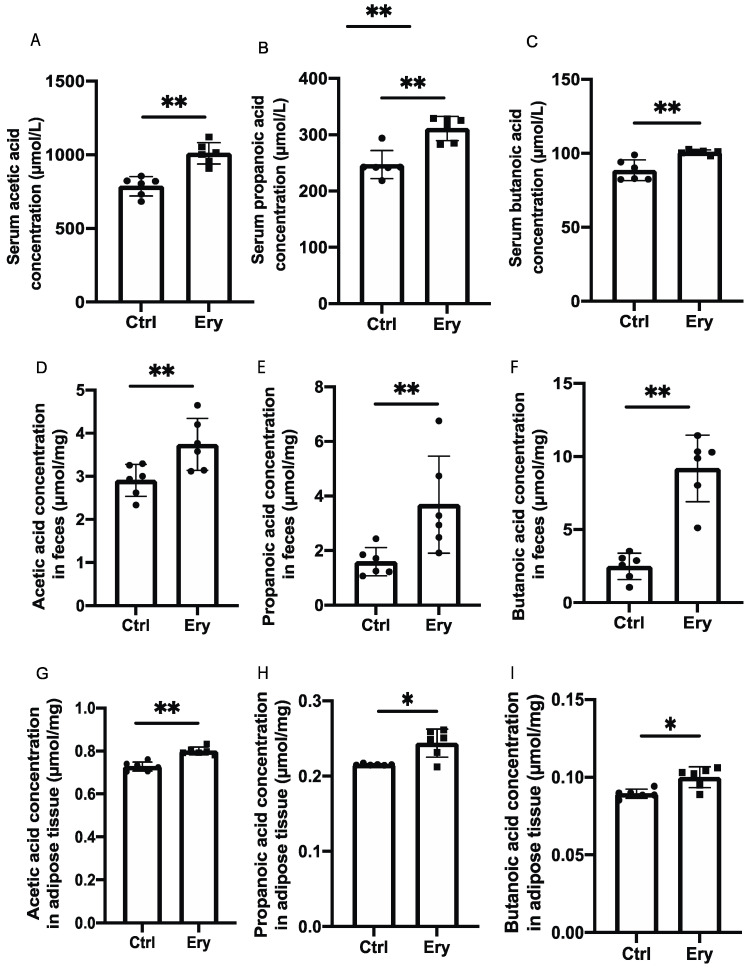
Erythritol administration increased short-chain fatty acid concentrations in the serum, feces, and white adipose tissue. (**A**) Serum acetic acid concentration (μmol/L) (*n* = 6). (**B**) Serum propanoic acid concentration (μmol/L) (*n* = 6). (**C**) Serum acetic acid concentration (μmol/L) (*n* = 6). (**D**) Acetic acid concentration in feces (μmol/mg) (*n* = 6). (**E**) Propanoic acid concentration in feces (μmol/mg) (*n* = 6). (**F**) Butanoic acid concentration in feces (μmol/mg) (*n* = 6). (**G**) Acetic acid concentration in white adipose tissue (μmol/mg) (*n* = 6). (**H**) Propanoic acid concentration in white adipose tissue (μmol/mg) (*n* = 6). (**I**) Butanoic acid concentration in white adipose tissue (μmol/mg) (*n* = 6). Data have been represented in terms of mean ± SD values. * *p* < 0.05, ** *p* < 0.01, using an unpaired *t*-test.

**Figure 5 ijms-22-05558-f005:**
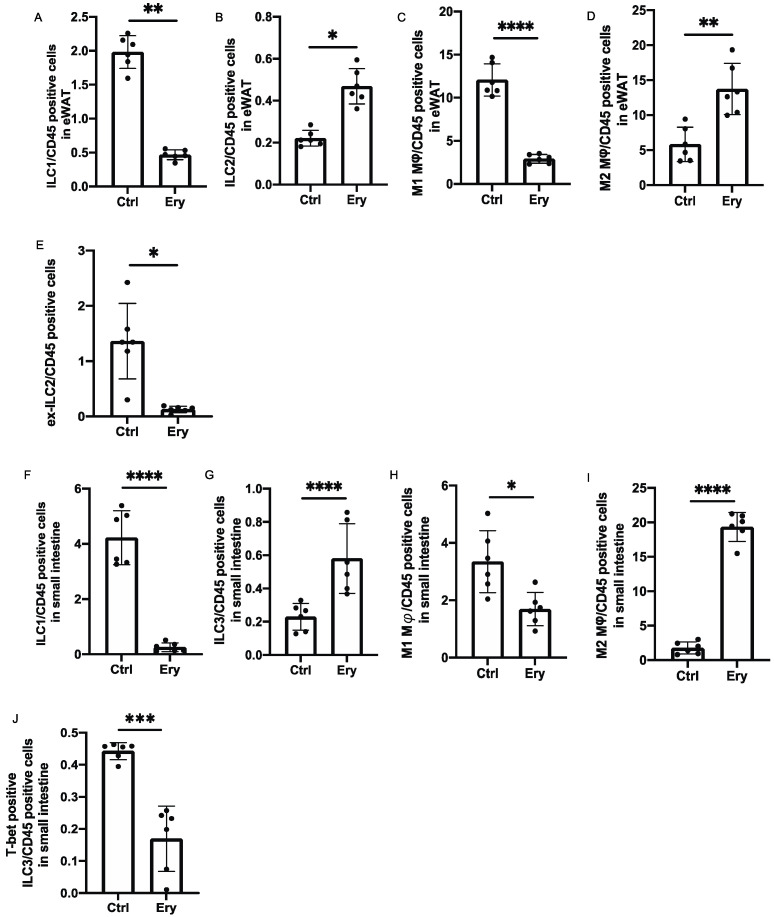
The Ery group had significantly lesser ILC1s and M1 macrophages in the white adipose tissue and lamina propria of the small intestine, ILC2s in the white adipose tissue, and ILC3s in the lamina propria of the small intestine than the Ctrl group. (**A**) ILC1:CD45-positive cell ratio in the epididymal white adipose tissue (eWAT) (*n* = 6). (**B**) ILC2:CD45-positive cell ratio in the eWAT (*n* = 6). (**C**) M1 macrophage:CD45-positive cell ratio in the eWAT (*n* = 6). (**D**) M2 macrophage:CD45-positive cell ratio in the eWAT (*n* = 6). (**E**) Ex-ILC2:CD45-positive cell ratio in the eWAT (*n* = 6). (**F**) ILC1:CD45-positive cell ratio in the lamina propria of the small intestine (*n* = 6). (**G**) ILC2:CD45-positive cell ratio in the lamina propria of the small intestine (*n* = 6). (**H**) M1 macrophage:CD45-positive cell ratio in the lamina propria of the small intestine (*n* = 6). (**I**) M2 macrophage:CD45-positive cell ratio in the eWAT (*n* = 6). (**J**) T-bet-positive ILC3:CD45-positive cell ratio in the eWAT (*n* = 6). Data have been represented in terms of mean ± SD values. * *p* < 0.05, ** *p* < 0.01, *** *p* < 0.001, and **** *p* < 0.0001, using an unpaired *t*-test.

**Figure 6 ijms-22-05558-f006:**
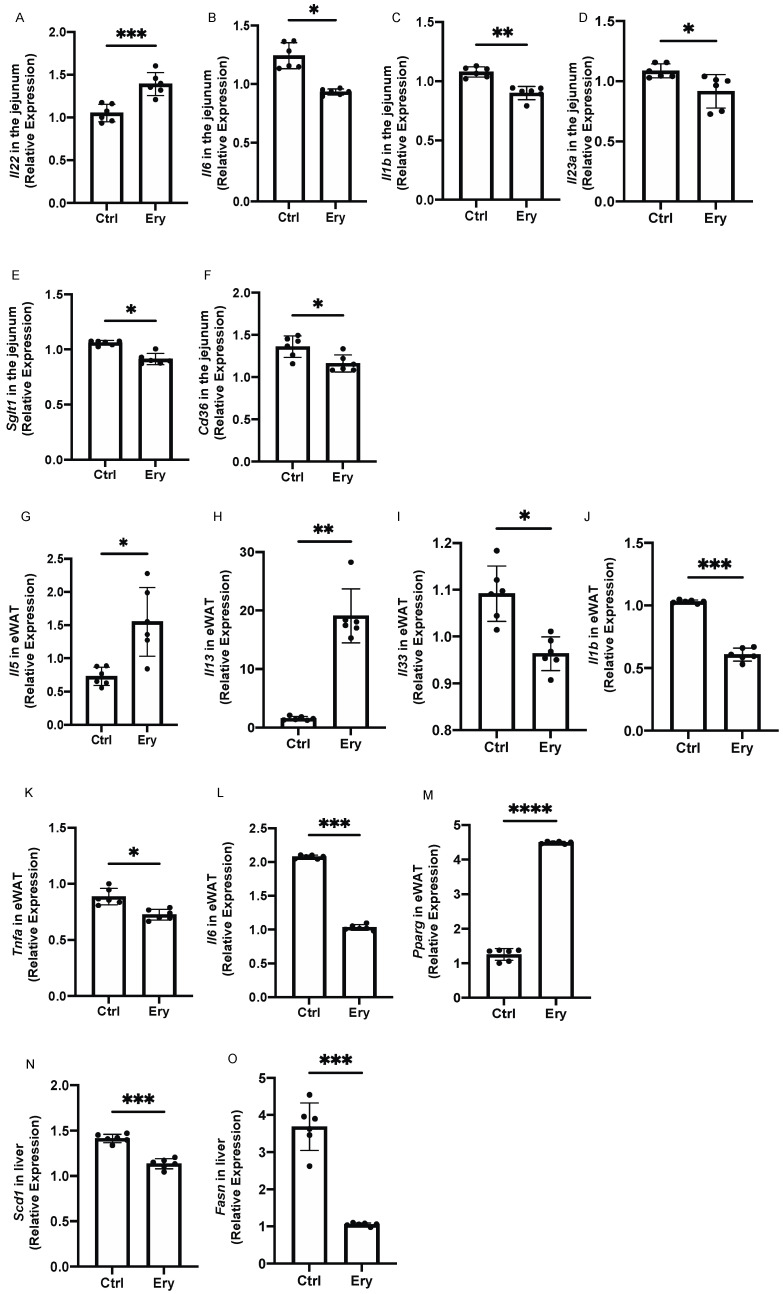
Erythritol decreased the expression of inflammation-related genes in the small intestine and white adipose tissue. (**A**) *Il22*, (**B**) *Il6*, (**C**) *Il1b*, (**D**) *Il23a,* (**E**) *Sglt1*, and (**F**) *Cd36* gene expression in the small intestine (*n* = 6). (**G**) *Il5*, (**H**) *Il13*, (**I**) *Il33*, (**J**) *Il1b*, (**K**) *Tnfa*, (**L**) *Il6,* and (**M**) *Pparg* gene expression in epididymal white adipose tissue (eWAT) (*n* = 6). (**N**) *Scd1* and (**O**) *Fasn* gene expression in liver (n = 6). Data have been represented in terms of mean ± SD values. * *p* < 0.05, ** *p* < 0.01, *** *p* < 0.001, and **** *p* < 0.0001 using an unpaired *t*-test.

**Figure 7 ijms-22-05558-f007:**
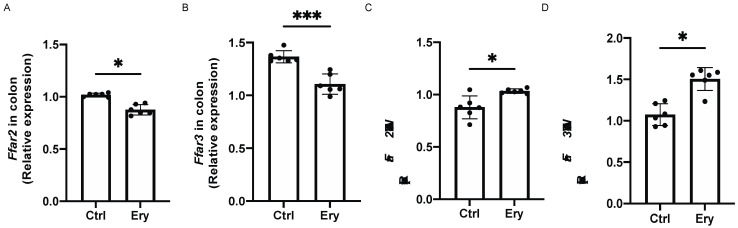
Erythritol decreased *Ffar2* and *Ffar3* expression in the colon and increased it in the eWAT. (**A**) *Ffar2* and (**B**) *Ffar3* gene expression in the colon (*n* = 6). (**C**) *Ffar2* and (**D**) *Ffar3* gene expression (*n* = 6) in the epididymal white adipose tissue (eWAT). Data have been represented in terms of mean ± SD values. * *p* < 0.05, ** *p* < 0.01, and *** *p* < 0.001, using an unpaired *t*-test.

## Data Availability

Data available on request due to restrictions e.g., privacy or ethical.

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
