# Peer review of "Erythritol Ameliorates Small Intestinal Inflammation Induced by High-Fat Diets and Improves Glucose Tolerance"

_ijms, 2021, doi:10.3390/ijms22115558_

Round 1

Reviewer 1 Report

Comments on

Erythritol ameliorates small intestinal inflammation induced by high-fat diets and improves glucose tolerance

The authors are evaluating the impact of a sugar alcohol, erythritol, on body weight gain, energy expenditure and glucose metabolism of mice fed with a high fat diet. In addition, the authors tried to link the beneficial effects of erythritol on glucose metabolism and the modulation of intestinal innate immune cells and the reduction of intestinal inflammation. The authors conclude that erythritol treatment protected the mice from diet induced obesity and metabolic disturbances via the modulation of small intestinal immune cells. Although the current question is relevant and merit to be investigated, the observed effects of erythritol treatment on body weight gain, energy expenditure and glucose metabolism are very modest. Also, the statements or conclusions are not supported by the current results and the study requires further investigations to demonstrate the beneficial effects of erythritol on host metabolism.

Comments:

  • The number of animals used in the current study is low (n= 6). The minimal number of animals advised to be used in metabolism studies is 8 animal per condition as the response of the animals to HFD feeding can vary a lot between animals. The number used in this study is not enough and might lead to a type 1 error. Also, the authors should present all the data using a scatter plot with bar to show the distribution.
  • Inappropriate statistical analysis: According to the legends, the authors performed a paired t-test instead of using an unpaired t-test. The paired t-test analysis is valid for the VO2 and VCO2 results in figure 1 I and K but not for the rest. Also, the authors wrote in the materials and methods section that the statistical analyses used in the study are the parametric and non-parametric t-tests. Can the authors comment on this?
  • The authors are making strong statements and conclusions regarding the effect of erythritol on body weight and energy expenditure. Also, the authors performed one single measurement of food intake during the protocol, which is not sufficient to conclude about the effect of erythritol on feeding behaviour of the animals. The food intake of the mice can vary over time and ideally, the authors should evaluate the feeding behaviour by calculating the cumulative food intake during the HFD feeding.
  • The results regarding the effects of erythritol on body weight and energy expenditure are not convincing. The graph 1H and 1J are showing similar VO2 consumption and VCO2 production between the groups. The curves are overlapping and do not show any significant differences. I have some concerns regarding the results in figures 1 I and 1K. Perhaps the authors should give more explanation about how they processed the data?
  • The authors should support their findings with additional analyses performed on interscapular brown adipose tissue (iBAT) and inguinal white adipose tissue (iWAT). The authors can analyse the gene expression of some thermogenic markers (e.g. Ucp-1, Prdm16, Pgc1a as well as Tmem26 and Tbx1(beiging markers)) as well as performing some histology analysis to analyse the overall structure of iBAT and iWAT.
  • The impact of erythritol on insulin resistance of mice fed with the HFD is not convincing and the AUC calculations for insulin resistance test seems to be wrong. The AUC graph in 1G looks similar to the one in 1E. The SD values in 1F are important in the control group and cannot correspond to the AUC values calculated in 1G. Can the authors comment on this? Also, the glucose metabolism data should be supported by the analysis of insulin levels.
  • The authors state that they have a trend in decreasing the weight of liver and epididymal fat. However, the results in figure 2A-2D are showing no trend at all. Also, this do not support their findings regarding the impact of erythritol on adipose tissue structure and adipocyte size. The authors should support these findings by performing some gene expression analysis on hepatic and adipose tissue lipid metabolisms as well as leptin level quantification in the blood. Also, scale bars should be added to the panels of figure 3A.
  • The values of SCFA (Figure 4A-4C) measured in sera samples are impressively high. Usually, HFD feeding and obesity are associated with significant decreased levels of SCFA. Can the authors comment on this?
  • In figure 5, I have concerns regarding the significance of the applied statistical test. The standard deviation is very important in some results yet the statistical test is significant: Figure 5E, H and J. Can the authors comment on this? The authors should present the data using a scatter plot with bar.
  • The authors used the gene expression analysis to investigate the impact of erythritol on intestinal inflammation and glucose transporters. The impact of erythritol on the gene expression of glucose and lipid transporters as well as on interleukins is almost non-existing and not convincing. Also, the data do not add any value to the manuscript. Same for the analysis of Ffar2 and Ffar3 gene expression in figure 7. The gene expression analysis of Ffar2 and Ffar3 do not give any relevance to their biological impact on host metabolism. In order to evaluate the ability of erythritol to improve host metabolism via SCFA production it is more suitable to use a knockout model.
  • Can also the authors state in the manuscript the reason for choosing only the jejunal tissue to perform the gene expression as well as the morphometric analyses?

The introduction section:

- References 7 and 8 are studies performed on rats. The authors used the term “healthy subjects” which might confuse the reader and suggest that the studies were performed on human subjects. Another term might be used instead of “healthy subjects”.

- The authors should modify the introduction. Perhaps less information regarding the description of erythritol is needed (line 46 to 54) and the authors should cite studies investigating the impact of erythritol on host physiology especially that two articles recently published have investigated the impact of erythritol on host metabolism (Hootman KC. 2017. PNAS and Dayoun Lee, 2020, Molecules). Also, a section regarding the link between obesity, adipose tissue and intestinal inflammation is missing. This might help the reader to understand the impact of obesity on ILC2 and ILC3 cells as well as the impact of innate immune cell dysregulation and the development of the so-called low-grade chronic inflammation and metabolic disturbances under high-fat feeding.

The results section:

The title of figure 2: The title needs to be more precise (line 189)

The title of figure 5 is wrong (line 352)

The discussion section:

  • The authors should discuss the following papers: Hootman KC, 2017, PNAS and Dayoun Lee, 2020, Molecules. The authors should try to discuss the relevance of their finding regarding human physiology, especially that Erythritol have been associated with weight gain in humans (Hootman KC, 2017, PNAS).
  • Line 466 to line 470: the authors state that Ffar2 expression is increased under HFD to suppress inflammation. In the current study the authors showed the opposite effect under erythritol treatment, which suggests according to authors discussion that erythritol might increase intestinal inflammation by reducing Ffar2 expression. The authors should rephrase this part of the discussion to make it clearer for the reader.
  • The sections on SGLT1 and CD36 should be modified according to the comments above.
  • Line 504-507: the authors wrote that erythritol treatment increased SCFA production, which act on FFAR2 to reduce intestinal inflammation. The authors should buffer the statement and rephrase their conclusion as the data do not support this statement. The current findings are only associations and no direct causalities have been demonstrated.

Author Response

Responses to Reviewer 1 1. The number of animals used in the current study is low (n= 6). The minimal number of animals advised to be used in metabolism studies is 8 animal per condition as the response of the animals to HFD feeding can vary a lot between animals. The number used in this study is not enough and might lead to a type 1 error. Also, the authors should present all the data using a scatter plot with bar to show the distribution.   Response Thank you for your valuable suggestion. It was not possible to implement case settings in advance, and from the perspective of animal welfare, it was desirable to conduct experiments with six mice per cage per group. In fact, we were able to detect a significant difference with n=6, so we believe that the number of cases was sufficient. In addition, we have modified the figures to scatter plots.     2. Inappropriate statistical analysis: According to the legends, the authors performed a paired t-test instead of using an unpaired t-test. The paired t-test analysis is valid for the VO2 and VCO2 results in figure 1 I and K but not for the rest. Also, the authors wrote in the materials and methods section that the statistical analyses used in the study are the parametric and non-parametric t-tests. Can the authors comment on this?   Response Thank you for your valuable comments. As you say, statistical analysis was incorrectly stated, the correct analysis was unpaired t-test. We have modified the sentences in the Materials and methods section described as below.   Materials and methods “Differences between two groups were assessed using the unpaired t-test and the paired t-test for the VO2 and VCO2, respectively.”   3. The authors are making strong statements and conclusions regarding the effect of erythritol on body weight and energy expenditure. Also, the authors performed one single measurement of food intake during the protocol, which is not sufficient to conclude about the effect of erythritol on feeding behaviour of the animals. The food intake of the mice can vary over time and ideally, the authors should evaluate the feeding behaviour by calculating the cumulative food intake during the HFD feeding.   Response Thank you for your valuable comments. According to your comments, we have added a figure of the cumulative food intake instead of the figure of food intake at 20 weeks.                                       Materials and methods “The cumulative oral intake was measured for 12 weeks, and water consumption of the mice were measured at 20 weeks.”   Results ““In contrast, the cumulative oral intake and water consumption did not significantly differ between both groups (Fig. 1B and 1C).” ”   4. The results regarding the effects of erythritol on body weight and energy expenditure are not convincing. The graph 1H and 1J are showing similar VO2 consumption and VCO2 production between the groups. The curves are overlapping and do not show any significant differences. I have some concerns regarding the results in figures 1 I and 1K. Perhaps the authors should give more explanation about how they processed the data?   Response Thank you for your valuable comments. The O2/CO2 analyzer we used in this study measures the oxygen consumption rate (VO2) and carbon dioxide production rate (VCO2) in the cage every 3 minutes, which can obtain 20 pieces of data per hour. In each group of 6 animals, the mean and SD values of the light and dark periods were calculated. Hence, the data is very accurate, and although there appears to be no difference in the line graph, when compared using the above method, significant differences appear as shown in the bar graph. As you say, we have added the detail of O2/CO2 analyzer in the Materials and methods section described as below.   Materials and methods “The O2/CO2 analyzer we used in this study measures the VO2 and VCO2 in the cage every 3 minutes, which can obtain 20 pieces of data per hour. In each group of 6 animals, the mean and SD values of the light and dark periods were calculated.”     5. The authors should support their findings with additional analyses performed on interscapular brown adipose tissue (iBAT) and inguinal white adipose tissue (iWAT). The authors can analyse the gene expression of some thermogenic markers (e.g. Ucp-1, Prdm16, Pgc1a as well as Tmem26 and Tbx1(beiging markers)) as well as performing some histology analysis to analyse the overall structure of iBAT and iWAT.   Response Thank you for your valuable comments. We agree that you pointed out, however, in this study, we focused on SCFA and FFAR2 and 3, and paid attention to eWAT, which strongly expresses FFAR2. It has been reported that brown fat and subcutaneous fat have weak expression of FFAR2, and we did not examine them in this study. In order to clarify the purpose of our study, we have added the sentence in the introduction according to your suggestion.   Introduction   In this study, we focus on SCFA and FFAR2 and FFAR3, with a particular focus on eWAT, which strongly expresses FFAR2, and erythritol was administered to experimental animals to evaluate its protective effect against metabolic disorders caused by a high-fat diet (HFD), while focusing on intestinal innate immunity.     6. The impact of erythritol on insulin resistance of mice fed with the HFD is not convincing and the AUC calculations for insulin resistance test seems to be wrong. The AUC graph in 1G looks similar to the one in 1E. The SD values in 1F are important in the control group and cannot correspond to the AUC values calculated in 1G. Can the authors comment on this? Also, the glucose metabolism data should be supported by the analysis of insulin levels.   Response Thank you for your kind comments. As you say, we have modified Figure 1G.    Unfortunately, however, we did not have the data of insulin. Therefore, we have added the sentences in the limitation described as below.   Discussion Moreover, we did not have the data of insulin in the iPGTT and ITT.     7. The authors state that they have a trend in decreasing the weight of liver and epididymal fat. However, the results in figure 2A-2D are showing no trend at all. Also, this do not support their findings regarding the impact of erythritol on adipose tissue structure and adipocyte size. The authors should support these findings by performing some gene expression analysis on hepatic and adipose tissue lipid metabolisms as well as leptin level quantification in the blood. Also, scale bars should be added to the panels of figure 3A.   Response Thank you for your valuable suggestion. As you say, we have modified the sentences described as below. According to your comment, we have added the figure of Pparg expression in the eWAT and Scd1 and Fasn expression in the liver as Figure 6M, N, and O. In addition, it is well known that leptin is a satiety hormone and SCFAs stimulate FFAR, which promotes the secretion of leptin from adipose tissue. In this study, the experimental animals were free-fed, but we did not measure appetite-related hormones because there was no apparent difference in food intake between the two groups. Therefore, we have added the sentences described as below in the Results section. In addition, we have added the scale bar into the figure 3A.   Results “The liver and epididymal fat weights in the Ery group did not significantly differ, compared with those in the Ctrl group (Fig. 2A-D).”   “2.10. Erythritol promoted adipocyte differentiation and lipid metabolism in the liver The expression of Pparg, which is involved in adipocyte differentiation, was significantly up-regulated in the Ery group (p < 0.001; Fig. 6M), and the expression of enzymes involved in hepatic lipid metabolism, such as Scd1 and Fasn, was significantly down-regulated in the Ery group (Scd1: p < 0.001, Fasn: p < 0.001; Fig. 6N and 6O).”   Discussion “In this study, Pparg mRNA expression, which is involved in the differentiation of adipocytes in the WAT of the Ery group significantly increased, compared to that of Ctrl group. In the Ery group, the adipocytes became smaller, indicating the genetic changes. In a previous report, it was reported that mice with adipose tissue-specific knockout (KO) of Pparγ did not show any effect of short-chain fatty acid administration, such as weight loss or increase in insulin sensitivity, indicating that SCFA may be a selective PPARγ modulator [31]. In the present study, erythritol administration increased the concentration of short-chain fatty acids in WAT, which may have led to adipocyte miniaturization through increased Pparg mRNA expression.”   “Furthermore, erythritol treatment reduced Cd36-mediated absorption of saturated fatty acids, which decreased intrahepatic fat accumulation, significantly improved lipid profiles such as cholesterol, TG, and NEFA, and significantly decreased gene expression such as Scd1 and Fasn, which are involved in fatty acid synthesis in the liver It was also suggested that the expression of genes such as Scd1 and Fasn involved in fatty acid synthesis in the liver was also significantly decreased.”   Materials and methods “We performed real-time reverse transcription polymerase chain reaction (RT-PCR) to quantify the following: Il22, Il23a, Sglt1, and Cd36 mRNA expression levels in the small intestine; Il5, Il13, Il33, Il1b, Il6, Tnfa, and Pparg mRNA expression levels in the eWAT; Scd1 and Fasn mRNA expression levels in the liver; and Ffar2 and Ffar3 mRNA expression levels in the colon and eWAT. RT-PCR was performed using TaqMan Fast Advanced Master Mix (Applied Biosystems), according to the manufacturer’s instructions.” Figure legends “Scale bar: 100μm.”     8. The values of SCFA (Figure 4A-4C) measured in sera samples are impressively high. Usually, HFD feeding and obesity are associated with significant decreased levels of SCFA. Can the authors comment on this?   Response Thank you for your kind comments. As you say, the unit of serum SCFA concentration was wrong. We modified the unit from μmol/ml to μmol/L.   Figure legends “(A) Serum acetic acid concentration (μmol/L) (n = 6). (B) Serum propanoic acid concentration (μmol/L) (n = 6). (C) Serum acetic acid concentration (μmol/L) (n = 6).”     9. In figure 5, I have concerns regarding the significance of the applied statistical test. The standard deviation is very important in some results yet the statistical test is significant: Figure 5E, H and J. Can the authors comment on this? The authors should present the data using a scatter plot with bar.   Response Thank you for your valuable comments. According to your comments, we have changed the figures to scatter plots with bar.     10. The authors used the gene expression analysis to investigate the impact of erythritol on intestinal inflammation and glucose transporters. The impact of erythritol on the gene expression of glucose and lipid transporters as well as on interleukins is almost non-existing and not convincing. Also, the data do not add any value to the manuscript. Same for the analysis of Ffar2 and Ffar3 gene expression in figure 7. The gene expression analysis of Ffar2 and Ffar3 do not give any relevance to their biological impact on host metabolism. In order to evaluate the ability of erythritol to improve host metabolism via SCFA production it is more suitable to use a knockout model.   Response Thank you for your valuable comments. As you say, in oreder to observe the effect of SCFA, it is desirable to study with FFAR KO mice. On the other hand, our scope in this study is to elucidate the health effects of erythritol. As a result, we found that part of the health effect of erythritol is due to SCFA. Therefore, according to your comments, we have added the sentences described as below in the Discussion section.   Discussion “Whether the health effects of erythritol are SCFA-dependent will require further validation in FFAR KO mice.”     11. Can also the authors state in the manuscript the reason for choosing only the jejunal tissue to perform the gene expression as well as the morphometric analyses?   Response Thank you for your valuable comments. When using our kit for extracting mononuclear cells from LPL of the small intestine, only the jejunum was used because using the whole small intestine resulted in more impurities even using the density gradient centrifugation method with Percoll. Therefore, the jejunum was also used in RT-PCR and histology. We have modified the sentences in the Materials and methods section described as below.   Materials and methods “Lamina propria mononuclear cells of small intestine (mainly jejunum) were isolated using the Lamina Propria Dissociation Kit (130-097-410; Miltenyi Biotec, Germany), as per the manufacturer’s instructions.”     12. The introduction section: - References 7 and 8 are studies performed on rats. The authors used the term “healthy subjects” which might confuse the reader and suggest that the studies were performed on human subjects. Another term might be used instead of “healthy subjects”.   Response Thank you for your kind comments. We have modified the sentences in the Introduction section described as below.   Introduction “Blood glucose levels do not increase when erythritol is ingested by healthy animals; therefore, the blood insulin concentration is also unaffected [7,8].”     - The authors should modify the introduction. Perhaps less information regarding the description of erythritol is needed (line 46 to 54) and the authors should cite studies investigating the impact of erythritol on host physiology especially that two articles recently published have investigated the impact of erythritol on host metabolism (Hootman KC. 2017. PNAS and Dayoun Lee, 2020, Molecules). Also, a section regarding the link between obesity, adipose tissue and intestinal inflammation is missing. This might help the reader to understand the impact of obesity on ILC2 and ILC3 cells as well as the impact of innate immune cell dysregulation and the development of the so-called low-grade chronic inflammation and metabolic disturbances under high-fat feeding.   Response Thank you for your kind suggestion. According to your comments, we have modified the Introduction section described as below.   Introduction “A previous animal study demonstrated that the administration of erythritol improved fasting blood glucose levels and HOMA-IR [a].” “On the other hand, in a previous report, it was shown that intestinal immune cells remotely control chronic inflammation in adipose tissue and modulate systemic insulin sensitivity in obese mice fed a high-fat diet [b], which suggest that the link between intestinal and adipose tissue immunity plays a key role in the pathogenesis of obesity.”   References a. Lee, D.; Han, Y.; Kwon, E.Y.; Choi, M.S. D-Allulose Ameliorates Metabolic Dysfunction in C57BL/KsJ-Db/Db Mice. Molecules 2020, 25, doi:10.3390/molecules25163656. b. Kawano, Y.; Nakae, J.; Watanabe, N.; Kikuchi, T.; Tateya, S.; Tamori, Y.; Kaneko, M.; Abe, T.; Onodera, M.; Itoh, H. Colonic Pro-Inflammatory Macrophages Cause Insulin Resistance in an Intestinal Ccl2/Ccr2-Dependent Manner. Cell Metabolism 2016, 24, 295–310, doi:10.1016/j.cmet.2016.07.009.     13. The results section: The title of figure 2: The title needs to be more precise (line 189) The title of figure 5 is wrong (line 352)   Response Thank you for your kind comments. According to your comments, we have modified the titles of figures described as below.   “Figure 2. The weight of organs and blood biochemistry test results. Erythritol decreased hepatic enzyme and improved serum lipid levels.” “Figure 5. The Ery group had significantly lesser ILC1s and M1 macrophages in white adipose tissue and lamina propria of the small intestine, and significantly more ILC2s in white adipose tissue and ILC3s in lamina propria of the small intestine than the Ctrl group.”     14. The discussion section: • The authors should discuss the following papers: Hootman KC, 2017, PNAS and Dayoun Lee, 2020, Molecules. The authors should try to discuss the relevance of their finding regarding human physiology, especially that Erythritol have been associated with weight gain in humans (Hootman KC, 2017, PNAS).   Response Thank you for your valuable comments. According to your comments, we have added the sentence in the Discussion section described as below.   Discussion “On the other hand, in a human study, young adults with incident central adiposity gain had statistically significantly higher blood erythritol levels [a]. However, the authors note that this may involve endogenous production of erythritol from glucose via the pentose-phosphate pathway, which does not imply that erythritol is involved in the worsening of obesity.”   References a. Hootman, K.C.; Trezzi, J.P.; Kraemer, L.; Burwell, L.S.; Dong, X.; Guertin, K.A.; Jaeger, C.; Stover, P.J.; Hiller, K.; Cassano, P.A. Erythritol Is a Pentose-Phosphate Pathway Metabolite and Associated with Adiposity Gain in Young Adults. Proceedings of the National Academy of Sciences of the United States of America 2017, 114, E4233–E4240, doi:10.1073/pnas.1620079114.     • Line 466 to line 470: the authors state that Ffar2 expression is increased under HFD to suppress inflammation. In the current study the authors showed the opposite effect under erythritol treatment, which suggests according to authors discussion that erythritol might increase intestinal inflammation by reducing Ffar2 expression. The authors should rephrase this part of the discussion to make it clearer for the reader.   Response Thank you for your valuable comments. In a previous report, Lu et al [a]. reported that feeding a HFD increased the gene expression of FFAR2 and FFAR3 in the colon, compared to feeding a ND and the administration of short chain fatty acid decreased the expression of FFAR2 and FFAR3 in the colon. In addition, a positive correlation with the expression of PYY and GLP-1 has been reported in the same report, which suggested that high expression of FFAR2 and FFAR3 in the colon might prevent weight gain by increasing PYY and GLP-1 expression and suppressing food intake. In this study, erythritol-induced increase of short-chain fatty acids in the colon may have decreased the expression of FFAR2 and FFAR3 in the colon. Therefore, we have added the sentences described as below in the Discussion section.   Discussion “Furthermore, Lu et al [a]. reported that feeding a HFD increased the gene expression of FFAR2 and FFAR3 in the colon, compared to feeding a ND and the administration of short chain fatty acid decreased the expression of FFAR2 and FFAR3 in the colon. In addition, a positive correlation with the expression of PYY and GLP-1 has been reported in the same report, which suggested that high expression of FFAR2 and FFAR3 in the colon might prevent weight gain by increasing PYY and GLP-1 expression and suppressing food intake. In this study, erythritol-induced increase of short-chain fatty acids in the colon may have decreased the expression of FFAR2 and FFAR3 in the colon.”   References a. Lu, Y.; Fan, C.; Li, P.; Lu, Y.; Chang, X.; Qi, K. Short Chain Fatty Acids Prevent High-Fat-Diet-Induced Obesity in Mice by Regulating g Protein-Coupled Receptors and Gut Microbiota. Scientific Reports 2016, 6, doi:10.1038/srep37589.     • The sections on SGLT1 and CD36 should be modified according to the comments above.   Response Thank you for your valuable suggestion. We have added the sentences described as below.   Discussion “As with FFAR, this finding needs to be tested in KO mice to determine whether erythritol has a glycemic effect via changes in SGLT-1 expression.”   “Furthermore, erythritol treatment reduced Cd36-mediated absorption of saturated fatty acids, which decreased intrahepatic fat accumulation, significantly improved lipid profiles such as cholesterol, TG, and NEFA, and significantly decreased gene expression such as Scd1 and Fasn, which are involved in fatty acid synthesis in the liver It was also suggested that the expression of genes such as Scd1 and Fasn involved in fatty acid synthesis in the liver was also significantly decreased.”   • Line 504-507: the authors wrote that erythritol treatment increased SCFA production, which act on FFAR2 to reduce intestinal inflammation. The authors should buffer the statement and rephrase their conclusion as the data do not support this statement. The current findings are only associations and no direct causalities have been demonstrated.   Response Thank you for your valuable comments. According to your comments, we have modified the sentence described as below.   Discussion “In summary, erythritol significantly decreased metabolic disorders, such as obesity, glucose intolerance, dyslipidemia, and fat accumulation in the liver, which were induced by an HFD”

Reviewer 2 Report

International Journal of Molecular Sciences
Original Article ijms-1210462 titled: “Erythritol ameliorates small intestinal
inflammation induced by high-fat diets and improves glucose tolerance" by
Rena Kawano et al.
The present article is appropriate to this Journal. The study provides numerous
observations that are of general interest. Excessive sugar in foods and drinks is a
problem to patients with diabetes and obesity and, therefore, finding substitutes for
sugar in diets is a line of research with increasing interest. The list of parameters
determined provides a complete understanding of the present manuscript. Overall, this
manuscript is well focused, nevertheless, several points raised by this reviewer should
be clarified.
MINOR POINTS
Results
1. Figure 1 shows that BW decreased after erythritol administration. However,
neither liver nor epididymal fat weights decreased (Figure 2). Did the weight of
any organ or tissue decrease to explain the erythritol-induced BW diminution?
Please, indicate it (at least in the text).
2. In the legend of Figure 2 and then, in the subsection 2.4., please indicate
“hepatic enzyme activity” instead of “hepatic enzyme” referring to the levels of
ALT and AST.
3. Please, in the subsection 2.8, indicate what Sglt1 means.
4. In Figure 6, the titles of the Y-axis are too long. Please, shorten them and then,
the explanations should be indicated in the legend.
Discussion
1. The names of SCFA should be the same in Results and in Discussion. The
authors should use the IUPAC name or the common name, not both.
2. In lines 461-462, the authors indicate that “stimulation of FFAR2 suppressed
insulin signaling in adipocytes because of inhibition of Akt phosphorylation. In
this study, erythritol administration significantly increased FFAR2 and FFAR3
expression in the WAT”. A suppression of insulin signaling could be understood
as a kind of insulin resistance, however erythritol increased FFAR2 in WAT and
also improved glucose tolerance. Please, explain it to clarify.
3. In the line 494, the authors introduce the concept of “RORt-positive ILC3s” for
the first time in the manuscript. Please, use and explain this concept in the
Introduction, as already made for the T-bet transcription factor.
4. One of the limitations of this study is the lack of a third group using another
substitute of sugar to demonstrate that the effects here observed are erythritol
specific. Therefore, this Referee thinks that a paragraph with the limitations of
this study should be included.
Materials and methods
1. Oral intake and water consumption were measured at 20 weeks of age only for
24 hours. In order to be useful these data, both parameters should be taken for
several days and then the AUCs should be calculated.
2. Please, explain iPGTT and ITT in more detail, that is: How many days before
the sacrifice did the iPGTT perform? And the ITT? This important to know in
order to discard that these techniques did not alter some of the determinations
carried out in the samples obtained at the sacrifice.

Author Response

Response to Reviewer 2
Results
1. Figure 1 shows that BW decreased after erythritol administration. However, neither liver nor epididymal fat weights decreased (Figure 2). Did the weight of any organ or tissue decrease to explain the erythritol-induced BW diminution? Please, indicate it (at least in the text).

Response
Thank you for your valuable comments. Liver and epididymal fat weights tended to be lower in the erythritol group, although the differences were not statistically significant. However, the administration of erythritol had metabolic effects, such as reduced fat accumulation in the liver and improved inflammation in visceral fat, which suggested that subcutaneous fat was also reduced, although not measured, resulting in body weight diminution. Therefore, we have added the sentences described as below in the Discussion section.

Discussion
“On the other hand, liver and epididymal fat weights tended to be lower in the erythritol group, although the differences were not statistically significant. However, the administration of erythritol had metabolic effects, such as reduced fat accumulation in the liver and improved inflammation in visceral fat, which suggested that subcutaneous fat was also reduced, although not measured, resulting in body weight diminution.”

2. In the legend of Figure 2 and then, in the subsection 2.4., please indicate “hepatic enzyme activity” instead of “hepatic enzyme” referring to the levels of ALT and AST.

Response
Thank you for your kind comment. According to your comment, we have modified the figure legend described as below.

“Figure 2. The weight of organs and blood biochemistry test results. Erythritol decreased hepatic enzyme and improved serum lipid levels.”

3. Please, in the subsection 2.8, indicate what Sglt1 means.

Response
Thank you for your valuable suggestion. According to your comment, we have added the sentences described as below in the subsection 2.8.

Results
“Moreover, the expression of Sglt1, a glucose transporter which expresses in the epithelium of the small intestine and proximal tubules of the kidney, in the Ery group was lower than that in the Ctrl group (p = 0.030; Fig. 6E), and the expression of Cd36, a long-chain fatty acid transporter, in the Ery group was lower than that in the Ctrl group (p = 0.013; Fig. 6F).”

4. In Figure 6, the titles of the Y-axis are too long. Please, shorten them and then, the explanations should be indicated in the legend.

Response
Thank you for your kind comments. According to your comments, we have modified.

Discussion
1. The names of SCFA should be the same in Results and in Discussion. The authors should use the IUPAC name or the common name, not both.

Response
Thank you for your kind comments. According to your comments, we have modified them.

2. In lines 461-462, the authors indicate that “stimulation of FFAR2 suppressed insulin signaling in adipocytes because of inhibition of Akt phosphorylation. In this study, erythritol administration significantly increased FFAR2 and FFAR3 expression in the WAT”. A suppression of insulin signaling could be understood as a kind of insulin resistance, however erythritol increased FFAR2 in WAT and also improved glucose tolerance. Please, explain it to clarify.

Response
Thank you for your valuable comments. According to your comments, we have modifed the sentences in the Discussion section described as below.

Discussion
In addition, in other previous studies, the expression of FFAR2 and FFAR3 in visceral fat of mice fed a high-fat diet was significantly lower than that of mice fed a normal diet [a]. Taken together, we hypothesized that the expression of FFAR2 and FFAR3 was significantly increased in high-fat diet-fed mice in a compensatory manner to improve insulin resistance in visceral fat. In this study, erythritol administration significantly increased FFAR2 and FFAR3 expression in the WAT. This suggests that erythritol improved insulin resistance in visceral fat caused by HFD.

References
a. Lu, Y.; Fan, C.; Li, P.; Lu, Y.; Chang, X.; Qi, K. Short Chain Fatty Acids Prevent High-Fat-Diet-Induced Obesity in Mice by Regulating g Protein-Coupled Receptors and Gut Microbiota. Scientific Reports 2016, 6, doi:10.1038/srep37589.

3. In the line 494, the authors introduce the concept of “RORt-positive ILC3s” for
the first time in the manuscript. Please, use and explain this concept in the
Introduction, as already made for the T-bet transcription factor.

Response
Thank you for your kind comments. According to your comments, we have added the sentence described as below in the Introduction section.

Introduction
As well as ILC2, ILC3s exhibit plasticity and the function of ILC3s is altered by the expression of the transcription factors RORγt and T-bet [17]. On stimulation with cytokines such as IL-12 and IL-18, ex-RORγt-positive ILC3s with T-bet-positive characteristics, that is, T-bet-positive ILC3s, increase and RORγt-positive ILC3s decrease, indicating that ILC3s can respond to environmental cues. A previous study showed that T-bet-positive ILC3s can produce IFN-γ and suppress IL-17 and IL-22 production [18]. Thus, T-bet-positive ILC3s exert a function similar to that of ILC1.

4. One of the limitations of this study is the lack of a third group using another substitute of sugar to demonstrate that the effects here observed are erythritol specific. Therefore, this Referee thinks that a paragraph with the limitations of this study should be included.

Response
Thank you for your valuable comments. According to your comments, we have added the sentences described in the Discussion section as below.

Discussion
“A limitation of this study is the lack of a third group using another substitute of sugar to demonstrate that the effects here observed are erythritol specific.”

Materials and methods
1. Oral intake and water consumption were measured at 20 weeks of age only for 24 hours. In order to be useful these data, both parameters should be taken for several days and then the AUCs should be calculated.

Response
Thank you for your valuable comments. According to another reviewer’s comment, we have added the cumulative food intake. Unfortunately, however, we did not have the data of cumulative water intake. Therefore, we have added the figure of cumulative food intake.

2. Please, explain iPGTT and ITT in more detail, that is: How many days before the sacrifice did the iPGTT perform? And the ITT? This important to know in order to discard that these techniques did not alter some of the determinations carried out in the samples obtained at the sacrifice.

Response
Thank you for your valuable comments. According to your comments. We have added the sentences in the Materials and methods section described as below.

Materials and methods
“An iPGTT (2 g/kg body weight) was performed for 20-week-old mice that had been fasted for 16 h (6 animals per group, three days before sacrifice). In addition, after a 5 h fast, the mice were treated with 0.5 U/kg body weight insulin for the ITT (two days before sacrifice).”

Round 2

Reviewer 1 Report

The authors addressed all my comments. However, the authors wrote in the figure legends that they used a "paired t-test". The authors should correct this as they used an unpaired t-test for most of the statistical analyses. 

Line 361: an Italic form should be used for gene names